# Phase II Trial of CDX-3379 and Cetuximab in Recurrent/Metastatic, HPV-Negative, Cetuximab-Resistant Head and Neck Cancer

**DOI:** 10.3390/cancers14102355

**Published:** 2022-05-10

**Authors:** Julie E. Bauman, Ricklie Julian, Nabil F. Saba, Trisha M. Wise-Draper, Douglas R. Adkins, Paul O’Brien, Mary Jo Fidler, Michael K. Gibson, Umamaheswar Duvvuri, Margo Heath-Chiozzi, Diego Alvarado, Richard Gedrich, Philip Golden, Roger B. Cohen

**Affiliations:** 1Division of Hematology/Oncology, Department of Medicine, University of Arizona Cancer Center, University of Arizona, Tucson, AZ 85724, USA; rickliej@arizona.edu; 2Division of Hematology/Oncology, Department of Medicine, GW Cancer Center, The George Washington University, Washington, DC 20052, USA; 3Department of Hematology and Medical Oncology, Winship Cancer Institute, Emory University, Atlanta, GA 30322, USA; nfsaba@emory.edu; 4Division of Hematology/Oncology, Department of Medicine, University of Cincinnati Cancer Center, University of Cincinnati, Cincinnati, OH 45267, USA; wiseth@ucmail.uc.edu; 5Division of Hematology/Oncology, Department of Medicine, Siteman Cancer Center, Washington University School of Medicine, St. Louis, MO 63110, USA; dadkins@wustl.edu; 6Division of Hematology/Oncology, Department of Medicine, MUSC Hollings Cancer Center, Medical University of South Carolina, Charleston, SC 29425, USA; obrienpe@musc.edu; 7Department of Internal Medicine, Rush University Medical Center, Chicago, IL 60612, USA; mary_fidler@rush.edu; 8Division of Hematology, Department of Medicine, Vanderbilt-Ingram Cancer Center, Vanderbilt University Medical Center, Nashville, TN 37232, USA; mike.gibson.1@vumc.org; 9Division of Head and Neck Surgery, Department of Otolaryngology, UPMC Hillman Cancer Center, University of Pittsburgh Medical Center (UPMC), Pittsburgh, PA 15232, USA; duvvuriu@upmc.edu; 10Celldex Therapeutics, Hampton, NJ 08827, USA; mheathchiozzi@celldex.com (M.H.-C.); dalvarado@celldex.com (D.A.); rgedrich@celldex.com (R.G.); pgolden@celldex.com (P.G.); 11Division of Hematology/Oncology, Department of Medicine, Perelman School of Medicine, University of Pennsylvania, Philadelphia, PA 19104, USA; roger.cohen@pennmedicine.upenn.edu

**Keywords:** head and neck cancer, EGFR, ErbB3, cetuximab, CDX-3379

## Abstract

**Simple Summary:**

This phase II, Simon 2-stage, multicenter study evaluated the efficacy of the combination of CDX-3379 and cetuximab, monoclonal antibodies against ErbB3 and EGFR, respectively, in patients with recurrent/metastatic, HPV-negative, cetuximab-resistant head and neck cancer. The primary endpoint was overall response rate (ORR) in genomically unselected patients. Enhanced response was hypothesized in the *FAT1*-mutated cohort. The ORR in genomically unselected patients was 2/30 (6.7%), which did not meet criteria for further investigation. The overall response rate was 1/10 (complete response; 10%) in the *FAT1*-mutated versus 0/17 (0%) in the *FAT1*-wildtype cohorts. The most common AEs were diarrhea (83%) and acneiform dermatitis (53%), leading to dose modification in 21 patients (70%). The modest ORR coupled to clinically significant and dose-limiting toxicity preclude further development of this combination.

**Abstract:**

In phase I development, CDX-3379, an anti-ErbB3 monoclonal antibody, showed promising molecular and antitumor activity in head and neck squamous cell carcinoma (HNSCC), alone or in combination with cetuximab. Preliminary biomarker data raised the hypothesis of enhanced response in tumors harboring *FAT1* mutations. This phase II, multicenter trial used a Simon 2-stage design to investigate the efficacy of CDX-3379 and cetuximab in 30 patients with recurrent/metastatic, HPV-negative, cetuximab-resistant HNSCC. The primary endpoint was objective response rate (ORR). Secondary endpoints included ORR in patients with somatic *FAT1* mutations, progression-free survival (PFS), overall survival (OS), and safety. Thirty patients were enrolled from March 2018 to September 2020. The ORR in genomically unselected patients was 2/30 (6.7%; 95% confidence interval [CI], 0.8–22.1). Median PFS and OS were 2.2 (95% CI: 1.3–3.6) and 6.6 months (95% CI: 2.7–7.5), respectively. Tissue was available in 27 patients including one of two responders. ORR was 1/10 (complete response; 10%; 95% CI 0.30–44.5) in the *FAT1*-mutated versus 0/17 (0%; 95% CI: 0–19.5) in the *FAT1*-wildtype cohorts. Sixteen patients (53%) experienced treatment-related adverse events (AEs) ≥ grade 3. The most common AEs were diarrhea (83%) and acneiform dermatitis (53%). Dose modification was required in 21 patients (70%). The modest ORR coupled with excessive, dose-limiting toxicity of this combination precludes further clinical development. Dual ErbB3-EGFR inhibition remains of scientific interest in HPV-negative HNSCC. Should more tolerable combinations be identified, development in an earlier line of therapy and prospective evaluation of the *FAT1* hypothesis warrant consideration.

## 1. Introduction

Head and neck squamous cell carcinoma (HNSCC), arising from the mucosal linings of the oral cavity, pharynx, and larynx, represents the seventh most common cancer worldwide with more than 700,000 cases anticipated in 2021 [1]. In the U.S., the estimated number of new cases is 66,000 with approximately 14,000 projected deaths in 2021 [2]. Most patients with HNSCC present with locoregionally advanced, Stage III to IVb disease and are treated curatively with multimodality therapy. Treatment failure rates, however, remain high, with up to 60% and 30% of patients having local and distant treatment failure, respectively [3]. Among patients who develop recurrent/metastatic disease, overall survival (OS) remains poor and treatment options limited. Survival is especially poor in patients with recurrent/metastatic, human papilloma virus (HPV)-negative disease.

In HNSCC, overexpression of the epidermal growth factor receptor (EGFR), also known as HER1, has been detected in the majority of HPV-negative tumors and has been associated with reduced OS and increased risk of recurrence or metastasis [4,5]. Indeed, the recognition of EGFR as a negative prognostic biomarker propelled the development of cetuximab, a human/mouse chimeric monoclonal antibody (mAb) that binds the extracellular domain of EGFR, inhibiting downstream signaling. Cetuximab has received broad global regulatory approval for the treatment of locoregionally advanced HNSCC in combination with radiation therapy and of recurrent/metastatic HNSCC as a component of the EXTREME regimen [6,7]. In the U.S., cetuximab also has approval for single-agent use in patients with recurrent/metastatic HNSCC who have progressed following platinum-based therapy, based on a single-arm multicenter study in patients with recurrent/metastatic HNSCC who demonstrated a 13% objective response rate (ORR) [8]. The current first-line standard of care for patients with recurrent/metastatic HNSCC consists of the anti-programmed death receptor-1 (PD1) mAb, pembrolizumab, with or without platinum and 5-fluorouracil (5FU) doublet chemotherapy [9]. Standards of care for second-line therapy following failure of anti-PD1 mAb are evolving; cetuximab or cetuximab-based combinations are acceptable standards. Upon failure of anti-PD1 mAb, platinum, and cetuximab in the recurrent/metastatic setting, response rates dwindle and OS plummets. More effective therapy represents a major unmet need, especially for patients with HPV-negative disease.

The low response rate to cetuximab monotherapy reflects intrinsic or acquired resistance to EGFR inhibition in HPV-negative HNSCC, despite the high expression of the target. Preclinical work has identified multiple candidate resistance mechanisms that are potentially targetable [10]. Chief among these are accessory receptor tyrosine kinases that converge on similar downstream signaling cascades, including other members of the HER family (HER2 and HER3), the HGF/cMet pathway, and the JAK/STAT pathway. CDX-3379 is a human immunoglobulin G1 lambda mAb that specifically binds ErbB3 (HER3) at a unique epitope, locking ErbB3 in an auto-inhibited configuration and preventing ErbB3 from binding its ligand, neuregulin-1 (NRG1), or from dimerizing with EGFR and HER2. HNSCC expresses the highest levels of NRG1 compared to all solid tumor types, with HPV-negative tumors showing even higher expression than HPV-positive tumors [11]. Upregulation of ErbB3 signaling has been identified as a mechanism of acquired cetuximab resistance in HPV-negative HNSCC cell lines and xenograft models, including the novel mechanism of autocrine NRG1 secretion [12,13,14]. The co-administration of cetuximab and CDX-3379 significantly reduced tumor growth in both cetuximab-sensitive and cetuximab-resistant, HPV-negative HNSCC xenografts, disrupting this autocrine loop [11,12]. Genomic silencing of the tumor suppressor, *FAT1*, is associated with poor prognosis and cetuximab resistance [15,16,17]. *FAT1* knockout decreased phospho-EGFR, phospho-HER2, and phospho-ERK while upregulating total and phospho-ErbB3 protein levels, raising the hypothesis that somatic *FAT1* mutations may be a response biomarker for the combination of CDX-3379 and cetuximab [18]. These preclinical mechanistic studies stimulated the development of CDX-3379 in HNSCC, including cetuximab-resistant disease. Encouraging also was the molecular and clinical activity of CDX-3379 monotherapy in a phase I window trial in HNSCC, where ErbB3 activation was inhibited in 10 of 12 paired biopsies and measurable tumor regression was observed in 5 of 12 patients, including an exceptional partial response (PR) in a patient with HPV-negative oral cavity HNSCC harboring a *FAT1* mutation [19,20]. Intriguing anti-tumor activity was also observed in a phase 1b study evaluating the combination of CDX-3379 with other targeted therapies. In the cohort evaluating CDX-3379 plus cetuximab, a prolonged complete response (CR) was observed in a patient with recurrent/metastatic HNSCC who previously had progressed on cetuximab alone [21].

Here, we report results from a phase II trial evaluating CDX-3379 and cetuximab combination therapy in patients with recurrent/metastatic, HPV-negative HNSCC who had progressed on previous standard therapies including cetuximab. Results are presented for the total genomically unselected population as well as the cohort with retrospectively identified *FAT1* mutations, a candidate efficacy biomarker for CDX-3379.

## 2. Patients and Methods

### 2.1. Patients

This multicenter study was conducted at the University of Arizona Cancer Center (Tucson, AZ, USA), Emory University Winship Cancer Center (Atlanta, GA, USA), University of Cincinnati Cancer Institute (Cincinnati, OH, USA), Washington University Siteman Cancer Center (St. Louis, MO, USA), Medical University of South Carolina Hollings Cancer Center (Charleston, SC, USA), Rush University Medical Center (Chicago, IL, USA), Vanderbilt University Medical Center (Nashville, TN, USA), University of Pittsburgh Medical Center (Pittsburgh, PA, USA), and Penn Medicine Abramson Cancer Center (Philadelphia, PA, USA). The protocol was approved by all local institutional review boards, carried out in accordance with the Declaration of Helsinki and Good Clinical Practice, and registered with ClinicalTrials.gov (NCT03254927). All patients gave written informed consent. Primary inclusion criteria included: HPV-negative recurrent/metastatic HNSCC incurable with local treatment modalities; male or female; adult ≥ age 18 years; progression on systemic therapy in the recurrent/metastatic setting; clinical cetuximab resistance, defined as progression during or within 6 months of cetuximab exposure in the definitive or recurrent/metastatic setting; anti-PD1 mAb exposure unless a contraindication existed; Eastern Cooperative Oncology Group performance-status score of 0 to 1; and measurable disease according to the Response Evaluation Criteria in Solid Tumors (RECIST), version 1.1 [22]. Patients with nasal, paranasal sinus or nasopharyngeal WHO Type III carcinoma were excluded.

### 2.2. Study Design

This was a single-arm, non-blinded, phase II study conducted according to a Simon 2-stage design, in which one tumor response (CR or PR) was required in the first stage of 13 patients before completion of accrual to a total of 30 patients. The primary endpoint was ORR as assessed by local radiologic review according to RECIST 1.1, with the use of contrast-enhanced computed tomography or magnetic resonance imaging. Key secondary endpoints were clinical benefit rate (CBR; defined as CR, PR, or stable disease (SD) > 12 weeks), duration of response, progression-free survival (PFS), OS, and safety. Exploratory biomarkers, including mutation analysis by next-generation sequencing, were evaluated for their association with response. Adverse events (AEs) were graded by the treating investigator using the U.S. National Cancer Institute Common Terminology Criteria for Adverse Events, version 5.0. In accordance with the protocol, response-related endpoints were to be evaluated in patients who had received at least one dose of CDX-3379 and had at least one RECIST-measurable lesion at baseline. Patients who were not evaluable for the primary endpoint, e.g., attrition with no post-baseline RECIST assessment, were classified as non-responders and included in the denominator for the analysis of ORR.

Following completion of stage 1 of the study, where the observation of 1 CR met the criteria to proceed to stage 2, the study design was amended to include a cohort with somatic *FAT1* mutations. The rationale was based upon emerging genomic data across the three trials within the CDX-3379 development program that had enrolled patients with HNSCC: CDX3379-01 Phase 1b (CDX-3379 + cetuximab in refractory solid tumors) (*n* = 1); CDX3379-02 (CDX3379 monotherapy; window of opportunity study in HNSCC) (*n* = 3); and CDX3379-04 (the present phase II study of CDX-3379 + cetuximab) (*n* = 14 of the first 15 evaluable patients) [23]. Albeit a small retrospective patient dataset including 18 patients with available HNSCC tissue, mutations in *FAT1* (observed in 7/18 patients or 39%) and *NOTCH1*, 2, or 3 (observed in 10 of 18 patients or 56%) appeared to be associated with clinical activity of CDX-3379. Specifically, all 4 observed objective responses in the three above trials occurred in patients with *FAT1* mutations; 3 of the 4 responders also had co-existing *NOTCH* mutations [23]. Given these findings, the current study design was amended to enroll up to 45 total patients, specifying a cohort of 15 patients with *FAT1* mutations based on retrospective gene sequencing and assuming a prevalence of *FAT1* mutations of approximately 30% [24,25]. This amendment specified two hypothesis tests: the ORR in the genomically unselected cohort and the ORR in the *FAT1* mutation-positive cohort. In genomic all-comers, a sample size of 45 patients provided > 80% power to rule out an ORR < 20% based upon the lower 95% confidence bound for the underlying true response rate. In the *FAT1* mutation cohort, a sample size of 15 patients provided ≥ 78% power to rule out an ORR < 15% if the true ORR was ≥ 40%.

### 2.3. Treatment Plan

CDX-3379 was administered at a dose of 12 mg/kg IV on day 1 of each 21-day treatment cycle, with an option to increase the dose to 15 mg/kg if no treatment-related, grade > 1 AEs were observed during cycle 1. Cetuximab was administered at standard weekly dosing, including a 400 mg/m^2^ loading dose on cycle 1 day 1, followed by 250 mg/m^2^/week maintenance doses. In the case of attributable, serious, or intolerable AEs, two levels of dose reduction were available for CDX-3379 (10 mg/kg, 6 mg/kg) and cetuximab (200 mg/m^2^, 150 mg/m^2^). Treatment with CDX-3379 plus cetuximab continued until confirmation of progressive disease, the development of unacceptable AEs, or withdrawal of consent.

Tumor assessments were performed every 2 cycles (6 weeks) during treatment. Patients who discontinued treatment in the absence of progression continued to have assessments approximately every 12 weeks until documented progression or initiation of alternate anticancer therapies. Toxicity was continuously evaluated during treatment and for 30 days following the last dose of CDX-3379. Subsequently, patients were followed for survival, with contact every 12 weeks until study closure.

### 2.4. Biomarker Analyses

Candidate genomic biomarkers from baseline or archival formalin-fixed, paraffin-embedded biopsies were evaluated by retrospective next-generation sequencing (NGS) using a commercially available panel targeting > 1400 cancer-related genes (including *FAT1* and *NOTCH1-3*) for the first subset of samples (*n* = 14; Personalis, Menlo Park, CA, USA) and whole exome sequencing using the updated ImmunoID NeXT (Personalis, Menlo Park, CA, USA) platform for the remainder.

### 2.5. Statistical Analysis

ORR with exact 95% confidence intervals was calculated by the Clopper–Pearson method. Time-to-event endpoints were summarized with the use of Kaplan–Meier estimates and 95% confidence intervals. A data review team comprised of the steering committee (JEB, NFS, RBC) and the medical monitor (MH-C) continuously assessed safety and made recommendations to the sponsor regarding continuation of the trial.

## 3. Results

### 3.1. Patient Characteristics

A total of 30 patients (including 13 to stage 1 and 17 to stage 2) with previously treated HPV-negative HNSCC were enrolled from March 2018 to September 2020 and received at least one dose of CDX-3379. The flow of patients through the study is depicted in the CONSORT diagram (Figure 1).

The characteristics of the patients at baseline are summarized in Table 1. The median age was 62 years (range, 46 to 79). Twenty-three patients (76%) were current or former smokers. Patients had received a median of 3.5 previous lines of systemic anti-cancer therapy in the recurrent/metastatic setting, with 17 of 30 (57%) receiving ≥ 5. Eleven patients (37%) had received a cetuximab-containing regimen as their last treatment prior to study entry.

### 3.2. Protocol Treatment

As of the data cutoff date of September 30, 2020, one patient continued treatment and the remaining patients had discontinued protocol treatment. Disease progression (in 23 patients (77%)) and AEs regardless of attribution (in 4 (13%)) were the most common reasons for discontinuation.

#### 3.2.1. CDX-3379 Exposure

The mean number of CDX-3379 infusions per patient was 4.2 (range 1, 25) with a mean cumulative dose of 48.3 mg/kg (range, 12–276 mg/kg), representing a mean dose intensity of 3.7 mg/kg/week (range, 2–5 mg/kg/week). One patient (3%) was eligible for a dose increase of CDX-3379 to 15 mg/kg after cycle 1. Conversely, CDX-3379 dose reduction was required in 13 patients (43%).

#### 3.2.2. Cetuximab Exposure

The mean number of cetuximab infusions per patient was 10.8 (range 1, 62). The mean cumulative dose was 2549 mg/m^2^ (range 399–11,663 mg/m^2^), representing a dose intensity of 213 mg/m^2^/week (range, 124–261 mg/m^2^/week). Cetuximab dose reduction was necessary in 21 patients (70%).

### 3.3. Safety

Safety data, including treatment-emergent AEs attributed to CDX-3379 and/or cetuximab, are summarized in Table 2. A total of 28 patients (93%) reported AEs of any grade that were attributed to protocol treatment. The highest-grade AE regardless of attribution per individual patient was grade 3 in 19 patients (63%), grade 4 in 5 patients (17%), and grade 5 in 1 patient (3%). In order of prevalence, the most common AEs attributed to CDX-3379 were diarrhea (83%), hypomagnesemia (30%), hypokalemia (23%), and fatigue (23%). The most common AEs attributed to cetuximab were acneiform dermatitis (53%), hypomagnesemia (47%), diarrhea (43%), dry skin (23%), and fatigue (23%). The single non-progression-related death during treatment was due to respiratory arrest from mechanical airway obstruction during tracheostomy self-removal and cleaning and was not considered related to CDX-3379 or cetuximab.

The high rate of treatment-related AEs, including diarrhea prompting delay or dose reduction of CDX-3379 and/or cetuximab, was noted during stage 1. Therefore, a mitigation strategy was implemented by the safety monitoring committee when opening stage 2. First, prophylactic loperamide was administered prior to CDX-3379 and subsequent to cetuximab on day 1 of each cycle. In the event of diarrhea despite prophylaxis, patients were instructed to self-administer loperamide following each loose stool over the ensuing 48 h; should diarrhea persist, diphenoxylate-atropine sulfate was added. Despite high investigator and patient compliance with these measures, the rate of AEs requiring dose modification of CDX-3379 and/or cetuximab remained unacceptably high. In October 2020, the safety monitoring committee reviewed cumulative toxicity data. In light of only one additional objective response in the 17 patients enrolled to stage 2, the safety monitoring committee recommended closure of the study.

### 3.4. Efficacy

#### 3.4.1. ORR, Genomic All-Comers (*N* = 30)

Among all 30 enrolled patients, 1 (3%) experienced a CR and 1 (3%) a PR; thus, the ORR in genomically unselected patients was 6.7% (95% confidence interval [CI], 0.8–22.1). SD occurred in 12 of 30 patients (40%). Disease progression was the best overall response in 11 of 30 patients (53%). Five patients discontinued treatment prior to the first RECIST response assessment due to an AE, hospice enrollment, or death; as specified, they were classified as non-responders and included in the denominator for the calculation of ORR and other categorical efficacy parameters. The ORR of 6.7% was not considered worthy of further study in genomically unselected patients.

#### 3.4.2. ORR, *FAT1* Mutation-Positive Cohort (*N* = 10)

Targeted NGS of *FAT1* and *NOTCH* genes was completed in the available tumors from 12/13 patients accrued to stage 1 and 15/17 patients accrued to stage 2. Unfortunately, tissue was available from only 1 of the two confirmed responders, specifically the exceptional responder with CR lasting 18 months and not the responder with PR. Across stages 1 and 2, *FAT1* mutations were observed in 10 of 27 patients (37%), *NOTCH1* mutations in 8 (30%), *NOTCH2* mutations in 3 (11%), and *NOTCH3* mutations in 2 (7%). *FAT1* and/or *NOTCH* mutations were observed in 17 (63%) of patients. The ORR in the *FAT1*-mutated cohort was 1 of 10 patients (1 prolonged CR) or 10% (95% CI: 0.3, 44.5) versus 0% (95% CI: 0, 19.5) in the 17 patients without *FAT1* mutation. A < 15% or > 40% ORR among patients with somatic *FAT1* mutations could not be ruled out based on the 95% CI bounds. The exceptional responder had three additional pathologic somatic mutations: *NOTCH 1, NOTCH 2*, and *TP53*.

#### 3.4.3. Secondary Efficacy Endpoints

In the total study population, the disease control rate was 47% (95% CI 28.3, 65.7) and the CBR was 40% (95% CI 22.7, 59.4). The duration of response was 5 months and 18.7 months for the patients with PR and CR, respectively. The median follow-up for survival was 6.3 months (range, 1.1 to 18.4). The median PFS was 2.2 months (95% CI, 1.3 to 3.6) and median OS was 6.6 months (95% CI: 2.7, 7.5) as shown in Figure 2.

## 4. Discussion

The anti-ErbB3 mAb, CDX-3379, previously demonstrated encouraging albeit preliminary anti-tumor activity in HPV-negative HNSCC [19,20,21]. In the current phase II trial evaluating the combination of CDX-3379 and cetuximab in patients with recurrent/metastatic, HPV-negative, cetuximab-resistant HNSCC, the observed ORR was only 6.7%, or 2 of 30 patients. This ORR did not meet the criteria for further investigation in genomically unselected patients, in either the original (*n* = 30) or amended (*n* = 45) Simon 2-stage design, noting that the latter would have required accrual of 15 additional patients to realize optimal power. The toxicity of the regimen, particularly the targeted-mediated AEs of diarrhea and rash, appeared more intense than with either agent alone. This resulted in frequent dose reductions and suboptimal dose intensity of CDX-3379 in 43% of patients, despite the implementation of a mandatory prophylaxis strategy for diarrhea in stage 2. When the study reached the original sample size of *n* = 30, only 1 additional response had been observed; considering the toxicity of the regimen despite maximal medical measures, the safety monitoring committee recommended closure of the trial.

Due to strong preclinical rationale, the hypothesis that dual EGFR-ErbB3 blockade could improve oncologic outcomes relative to EGFR inhibition alone was previously investigated in a phase II randomized trial of duligotuzumab, a dual-action humanized IgG1 antibody that blocks ligand binding to EGFR and ErbB3, vs. cetuximab in patients with platinum-resistant, cetuximab-naïve, HPV-positive or -negative HNSCC [26]. Duligotuzumab demonstrated comparable, albeit not superior, activity to cetuximab in this population, either in biomarker-unselected or participants with NRG1-high tumors, and further development was abandoned. A peer-reviewed commentary suggested that the inclusion of HPV-positive and 100% cetuximab-naïve subjects may have obscured a meaningful efficacy signal, given that NRG1 over-expression is more prevalent in HPV-negative disease and is a major mechanism of acquired cetuximab-resistance [27]. Similar to the current trial, diarrhea was dose-limiting and appears to be a class effect of ErbB3 targeting; in contrast, the incidence of acneiform rash on the duligotuzumab arm was lower than for cetuximab alone. Given that the occurrence of acneiform rash remains the only established predictive biomarker for cetuximab efficacy in HNSCC, and may be associated with its immune mechanism of action [28], a fair question is whether the bi-specific antibody construct of duligotuzumab preserved the anti-tumor immune functions of cetuximab [29,30,31].

The occurrence of at least one exceptional responder in each of the three CDX-3379 trials enrolling patients with HNSCC, including the first stage of the present study, motivated a search for biomarkers that could enhance patient selection during subsequent development. *FAT1*, *NOTCH1*, *NOTCH2*, and *NOTCH3* were identified as genes of interest potentially associated with response to CDX-3379 [23]. The *FAT1* gene, located on chromosome 4q35, is the human ortholog of the Drosophila gene, ‘*Fat*’, and encodes the human atypical FAT1 transmembrane protein, which acts as a tumor suppressor. FAT1 is a cadherin-like protein that can bind β-catenin and antagonize its nuclear localization; such sequestration inhibits its transcriptional activation of Wnt target genes [32]. Loss of FAT1 function is thus associated with increased expression of the Wnt family of growth factors, which intersect with multiple important oncogenic pathways in HNSCC including EGFR and other ErbB family members, PI3K/Akt, and CDK4/6 [15,16,17]. In The Cancer Genome Atlas (TCGA) HNSCC databases, somatic mutations in *FAT1* were associated with lower *FAT1* gene expression and increased protein expression of HER3_PY1289. Corroborating an important role for FAT1 in aberrant ErbB signaling, *FAT1* knockout in HNSCC cell lines decreased phospho-EGFR, phospho-HER2, and phospho-ERK while upregulating total and phospho-ErbB3 protein levels [18]. Moreover, dual blockade of EGFR and ErbB3 was superior to EGFR inhibition alone in patient-derived HNSCC xenografts. In a single-institution cohort of 101 HNSCC patients, deleterious *FAT1* mutations were identified in 29% of surgical specimens; in both this cohort and the TCGA datasets, harboring a somatic *FAT1* mutation was a strong, independent, negative prognostic factor in patients with HPV-negative HNSCC [18,24]. Given the provocative findings from the exploratory biomarker analysis across the CDX-3379 program, the current trial was amended to better estimate ORR in the context of somatic *FAT1* mutation. While the ORR was 10% vs. 0% in the *FAT1*-mutated vs. *FAT1*-wildtype cohorts, respectively, this numerical difference was not statistically significant; neither was it considered clinically significant when considering the observed excess toxicity. We note that the amended analysis plan required 15 patients with *FAT1* mutations, and *FAT1* mutation status was not available in the second responder; power was therefore inadequate and cautious interpretation is warranted. An alternate, intriguing biomarker for anti-ErbB3 drugs would be activating *NRG1* fusions, found to be oncogenic drivers in approximately one-third of invasive mucinous adenocarcinomas of the lung, however only present in 0.5% of HNSCC, thus making them impractical for patient selection in HNSCC-specific trials [33].

In addition to dose delays and reductions decreasing the dose intensity of protocol therapy, the negative results may have been impacted by the resistant biology inherent to a heavily pretreated population. In this study, 97% of patients with recurrent/metastatic HNSCC had been previously treated with anti-PD1 checkpoint inhibitors, platinum chemotherapy, and cetuximab. Moreover, the majority (57%) had received ≥5 previous lines of systemic therapy in the recurrent/metastatic setting. Molecular mechanisms of acquired resistance to standard HNSCC treatments include upregulation of DNA repair, decreased influx/increased efflux of the drug, inhibition of apoptosis, and aberrant RTK signaling [34,35]. In this heavily pretreated setting, the latter three mechanisms may have contributed to the failure of CDX-3379 in combination with cetuximab to overcome clinical cetuximab resistance. Future studies should consider the development of ErbB3 and EGFR inhibitors in an earlier line of palliative therapy.

## 5. Conclusions

In summary, the modest ORR combined with the clinically significant and dose-limiting toxicity of CDX-3379 and cetuximab preclude further development of this combination. The amendment of the trial to include a hypothesis test for an improved ORR in a biomarker-enriched cohort, while a novel approach to the use of preliminary biomarker evidence, ultimately was underpowered when the study had to close for toxicity. Preclinical and clinical evidence continues to point towards a therapeutic role for dual ErbB3 and EGFR targeting in HPV-negative HNSCC; however, the successful development of both tolerable and effective drugs or combinations has been elusive. An intriguing development is an ErbB3-targeting antibody-drug conjugate, U3-1402, which has shown encouraging activity in ErbB3-overexpressing breast cancer and *EGFR*-mutant, EGFR inhibitor-resistant non-small cell lung cancer and would be of interest in combination with cetuximab in HNSCC [36,37]. Should more tolerable drug combinations be identified, development in an earlier line of treatment and prospective evaluation of the *FAT1* hypothesis warrant consideration.

## Figures and Tables

**Figure 1 cancers-14-02355-f001:**
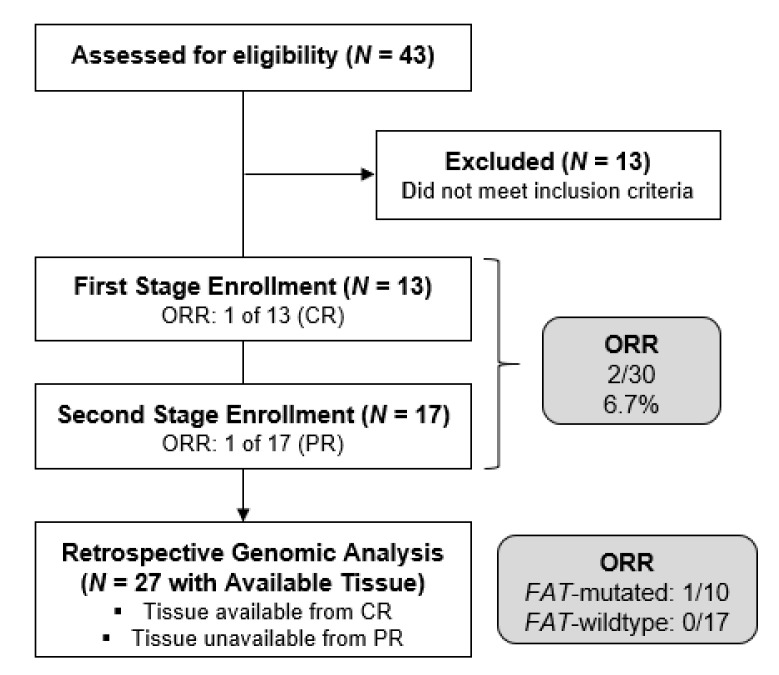
CONSORT diagram.

**Figure 2 cancers-14-02355-f002:**
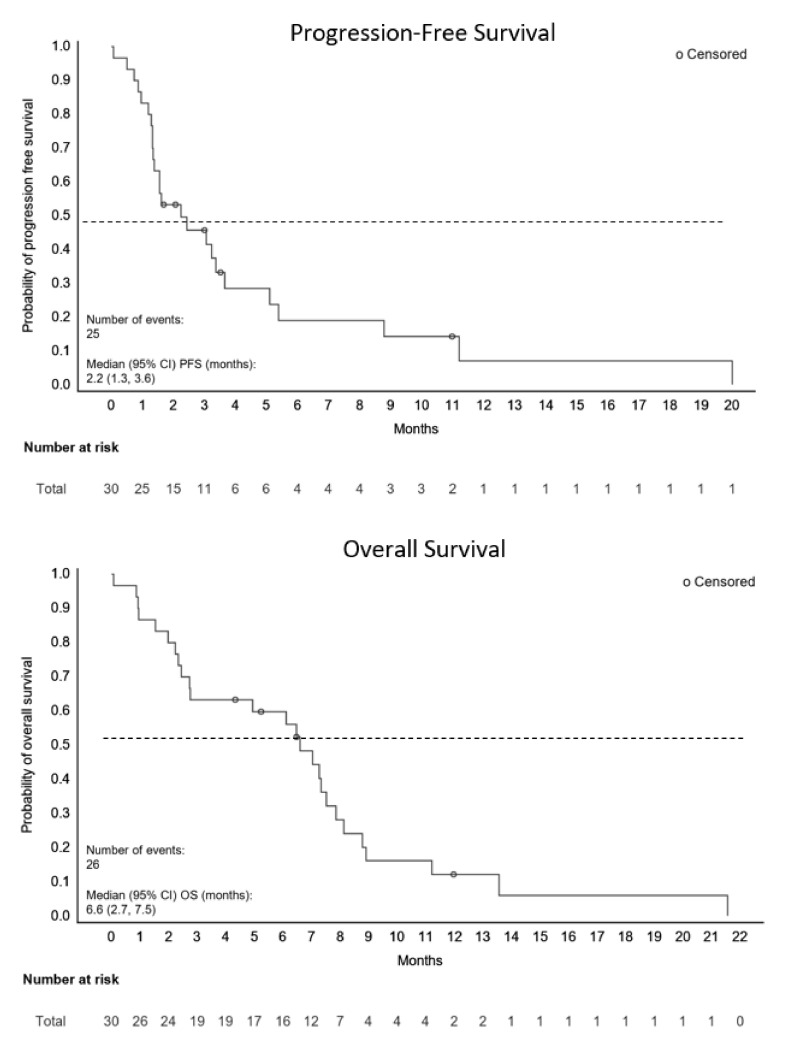
Progression-Free and Overall Survival.

**Table 1 cancers-14-02355-t001:** Baseline patient characteristics.

Characteristic	*N* = 30
Age, years	
Mean (SD)	62.0 (8.5)
Range (min, max)	46, 79
Sex, *N* (%)	
Male	26 (87)
Female	4 (13)
Race, *N* (%)	
White	25 (83)
Black or African American	3 (10)
Asian	1 (3)
Other	1 (3)
Ethnicity, *N* (%)	
Not Hispanic or Latino	26 (87)
Hispanic or Latino	4 (13)
Primary Tumor Site, *N* (%)	
Oral cavity	11 (37)
Oropharynx (HPV-negative)	8 (27)
Larynx	4 (13)
Hypopharynx	3 (10)
Nasopharynx (EBV-negative)	1 (3)
Other	3 (10)
ECOG Performance Status, *N* (%)	
0	4 (13)
1	26 (87)
Smoking status, *N* (%)	
Former	16 (53)
Current	7 (23)
Never	7 (23)
Subjects with any prior radiotherapy, *N* (%)	26 (87)
Subjects with any prior surgery, *N* (%)	30 (100)
Number of prior systemic therapy regimens in the recurrent/metastatic setting, *N* (%)	
1	0
2	1 (3)
3	4 (13)
4	6 (20)
≥5	17 (57)
Unknown	2 (17)
Prior systemic therapy, *N* (%)	
Cetuximab	29 (97)
Pembrolizumab	22 (73)
Carboplatin	21 (70)
Cisplatin	18 (60)

**Table 2 cancers-14-02355-t002:** Treatment-emergent adverse events related to protocol treatment.

MedDRA System Organ Class/ High Level Term/ Preferred Term	*N* = 30					
	Grade 1	Grade 2	Grade 3	Grade 4	Grade 5	Total
	*N* (%)	*N* (%)	*N* (%)	*N* (%)	*N* (%)	*N* (%)
Any AE related to CDX-3379	3 (10)	7 (23)	16 (53)	1 (3)	0	27 (90)
Gastrointestinal disorders	10 (33)	9 (30)	7 (23)	0	0	26 (87)
Diarrhea	12 (40)	7 (23)	6 (20)	0	0	25 (83)
Nausea and vomiting symptoms	5 (17)	1 (3)	0	0	0	6 (20)
Stomatitis	2 (7)	1 (3)	1 (3)	0	0	4 (13)
Oral dryness and saliva altered	2 (7)	0	0	0	0	2 (7)
Abdominal pain	1 (3)	0	0	0	0	1 (3)
Duodenal ulcer	0	0	1 (3)	0	0	1 (3)
Flatulence	1 (3)	0	0	0	0	1 (3)
Gastrointestinal disorder	0	1 (3)	0	0	0	1 (3)
Small intestinal hemorrhage	0	0	1 (3)	0	0	1 (3)
Metabolism and nutrition disorders	3 (10)	4 (13)	6 (20)	1 (3)	0	14 (47)
Hypomagnesemia	2 (7)	3 (10)	3 (10)	1 (3)	0	9 (30)
Hypokalemia	4 (13)	0	3 (10)	0	0	7 (23)
Decreased appetite	0	1 (3)	1 (3)	0	0	2 (7)
Dehydration	0	2 (7)	0	0	0	2 (7)
Hypophosphatemia	0	2 (7)	0	0	0	2 (7)
Hypoalbuminemia	0	1 (3)	0	0	0	1 (3)
Hypocalcemia	0	1 (3)	0	0	0	1 (3)
Skin and subcutaneous tissue disorders	2 (7)	4 (13)	7 (23)	0	0	13 (43)
Dermal and epidermal conditions NEC	4 (13)	1 (3)	1 (3)	0	0	6 (20)
Dermatitis acneiform	0	0	6 (20)	0	0	6 (20)
Rashes, eruptions and exanthems NEC	3 (10)	2 (7)	1 (3)	0	0	6 (20)
Pruritus	0	2 (7)	0	0	0	2 (7)
Alopecia	1 (3)	0	0	0	0	1 (3)
Onychoclasis	1 (3)	0	0	0	0	1 (3)
Palmar-plantar erythrodysesthesia syndrome	0	1 (3)	0	0	0	1 (3)
General disorders and administration site conditions	2 (7)	6 (20)	2 (7)	0	0	10 (33)
Asthenic conditions	2 (7)	5 (17)	0	0	0	7 (23)
Mucosal inflammation	0	0	2 (7)	0	0	2 (7)
Chills	0	1 (3)	0	0	0	1 (3)
Pain	1 (3)	0	0	0	0	1 (3)
Investigations	3 (10)	3 (10)	2 (7)	0	0	8 (27)
Electrocardiogram QT prolonged	1 (3)	1 (3)	1 (3)	0	0	3 (10)
Weight decreased	2 (7)	1 (3)	0	0	0	3 (10)
Digestive enzymes	0	2 (7)	0	0	0	2 (7)
Alanine aminotransferase increased	1 (3)	0	0	0	0	1 (3)
Blood creatinine increased	0	1 (3)	0	0	0	1 (3)
Lymphocyte count decreased	0	0	1 (3)	0	0	1 (3)
Eye disorders	2 (7)	1 (3)	0	0	0	3 (10)
Conjunctival hyperemia	0	1 (3)	0	0	0	1 (3)
Eye pruritus	1 (3)	0	0	0	0	1 (3)
Ocular discomfort	1 (3)	0	0	0	0	1 (3)
Vision blurred	1 (3)	0	0	0	0	1 (3)
Nervous system disorders	2 (7)	0	1 (3)	0	0	3 (10)
Dysgeusia	1 (3)	0	0	0	0	1 (3)
Encephalopathy	0	0	1 (3)	0	0	1 (3)
Headache	1 (3)	0	0	0	0	1 (3)
Peripheral sensory neuropathy	1 (3)	0	0	0	0	1 (3)
Injury	1 (3)	1 (3)	0	0	0	2 (7)
Radiation skin injury	0	1 (3)	0	0	0	1 (3)
Skin laceration	1 (3)	0	0	0	0	1 (3)
Infections and infestations	1 (3)	0	0	0	0	1 (3)
Fungal skin infection	1 (3)	0	0	0	0	1 (3)
Renal and urinary disorders	1 (3)	0	0	0	0	1 (3)
Proteinuria	1 (3)	0	0	0	0	1 (3)
Reproductive system and breast disorders	1 (3)	0	0	0	0	1 (3)
Perineal rash	1 (3)	0	0	0	0	1 (3)
**Any AE related to cetuximab**	**5 (17)**	**8 (27)**	**13 (43)**	**2 (7)**	**0**	**28 (93)**
Skin and subcutaneous tissue disorders	7 (23)	7 (23)	9 (30)	0	0	23 (77)
Dermatitis acneiform	5 (17)	3 (10)	8 (27)	0	0	16 (53)
Dermal and epidermal conditions NEC	6 (20)	1 (3)	1 (3)	0	0	8 (27)
Rashes, eruptions and exanthems NEC	3 (10)	2 (7)	1 (3)	0	0	6 (20)
Pruritus	1 (3)	2 (7)	0	0	0	3 (10)
Palmar-plantar erythrodysesthesia syndrome	2 (7)	0	0	0	0	2 (7)
Urticaria	2 (7)	0	0	0	0	2 (7)
Dermal cyst	0	1 (3)	0	0	0	1 (3)
Erythema	0	1 (3)	0	0	0	1 (3)
Onychoclasis	1 (3)	0	0	0	0	1 (3)
Metabolism and nutrition disorders	3 (10)	3 (10)	9 (30)	2 (7)	0	17 (57)
Magnesium metabolism disorders	4 (13)	4 (13)	5 (17)	2 (7)	0	15 (50)
Hypokalaemia	5 (17)	0	3 (10)	0	0	8 (27)
Hypophosphatemia	0	3 (10)	0	0	0	3 (10)
Dehydration	0	1 (3)	1 (3)	0	0	2 (7)
Hypocalcemia	1 (3)	1 (3)	0	0	0	2 (7)
Decreased appetite	0	1 (3)	0	0	0	1 (3)
Hypoalbuminemia	0	1 (3)	0	0	0	1 (3)
Gastrointestinal disorders	5 (17)	7 (23)	3 (10)	0	0	15 (50)
Diarrhea	5 (17)	7 (23)	1 (3)	0	0	13 (43)
Nausea and vomiting symptoms	4 (13)	1 (3)	0	0	0	5 (17)
Abdominal pain	1 (3)	0	0	0	0	1 (3)
Dysphagia	0	0	1 (3)	0	0	1 (3)
Lip dry	1 (3)	0	0	0	0	1 (3)
Small intestinal hemorrhage	0	0	1 (3)	0	0	1 (3)
Stomatitis	1 (3)	0	0	0	0	1 (3)
General disorders and administration site conditions	2 (7)	6 (20)	2 (7)	0	0	10 (33)
Asthenic conditions	2 (7)	5 (17)	0	0	0	7 (23)
Chills	1 (3)	1 (3)	0	0	0	2 (7)
Mucosal inflammation	0	0	2 (7)	0	0	2 (7)
Pain	1 (3)	0	0	0	0	1 (3)
Infections and infestations	4 (13)	4 (13)	0	0	0	8 (27)
Skin structures and soft tissue infections	2 (7)	4 (13)	0	0	0	6 (20)
Fungal infections NEC	3 (10)	1 (3)	0	0	0	4 (13)
Groin infection	1 (3)	0	0	0	0	1 (3)
Lip infection	1 (3)	0	0	0	0	1 (3)
Investigations	2 (7)	3 (10)	1 (3)	0	0	6 (20)
Digestive enzymes	0	2 (7)	0	0	0	2 (7)
Weight decreased	1 (3)	1 (3)	0	0	0	2 (7)
Alanine aminotransferase increased	1 (3)	0	0	0	0	1 (3)
Blood creatinine increased	0	1 (3)	0	0	0	1 (3)
Electrocardiogram QT prolonged	0	1 (3)	0	0	0	1 (3)
Lymphocyte count decreased	0	0	1 (3)	0	0	1 (3)
Eye disorders	1 (3)	1 (3)	1 (3)	0	0	3 (10)
Blepharitis	0	0	1 (3)	0	0	1 (3)
Conjunctival hyperemia	0	1 (3)	0	0	0	1 (3)
Ocular discomfort	1 (3)	0	0	0	0	1 (3)
Vision blurred	1 (3)	0	0	0	0	1 (3)
Nervous system disorders	3 (10)	0	0	0	0	3 (10)
Headache	2 (7)	0	0	0	0	2 (7)
Peripheral sensory neuropathy	1 (3)	0	0	0	0	1 (3)
Injury	1 (3)	1 (3)	0	0	0	2 (7)
Radiation skin injury	0	1 (3)	0	0	0	1 (3)
Skin laceration	1 (3)	0	0	0	0	1 (3)
Reproductive system and breast disorders	1 (3)	0	0	0	0	1 (3)
Perineal rash	1 (3)	0	0	0	0	1 (3)

Column header counts and denominators are the number of treated subjects. Subjects are count-ed at most once in each row and under the highest grade reported. High Level Term is omitted when there is only one associated Preferred Term. Treatment-emergent AEs are defined as events reported between first dose and 30 days after the last dose of study treatment. AEs are categorized using MedDRA version 23.1.

## Data Availability

Data supporting the reported results can be found at ClinicalTrials.gov, NCT03254927.

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
