# Peer review of "Phase II Trial of CDX-3379 and Cetuximab in Recurrent/Metastatic, HPV-Negative, Cetuximab-Resistant Head and Neck Cancer"

_cancers, 2022, doi:10.3390/cancers14102355_

Round 1

Reviewer 1 Report

In this study, Bauman et al., has evaluated the efficacy of the combination of CDX-3379 and cetuximab phase II, Simon 2-stage, multicenter study, in 30 HPV-negative head and neck cancer patients who are resistant to cetuximab monotherapy. Inclusion criteria and study design has made the study conclusions strong. Using the effective study design and adhering to the standards of Response Evaluation Criteria in Solid Tumors (RECIST) the authors were able to conclude without doubt that the modest objective response rate combined with the clinically significant and dose-limiting toxicity of CDX-3379 and cetuximab preclude further development of this combination.

In a previous study presented in the 2019 ASCO meeting, the authors have studied the efficacy of the same combination in advance HNSCC in a phase II trial(https://ascopubs.org/doi/abs/10.1200/JCO.2019.37.15_suppl.6025). The authors in that presentation concluded that CDX-3379 in combination with cetuximab is well tolerated with the primary toxicity of diarrhea. Also, it was claimed that antitumor activity was observed in cetuximab-resistant HNSCC patients, including an ongoing, durable complete response. The authors need to elaborate the present study with the findings of 2019 in the discussion section as both the study has the same clinical trial initiative NCT03254927.

The authors are advised to include a paragraph in the discussion why studies are to be performed and to consider development of ErbB3 and EGFR inhibition in an earlier line of palliative therapy. The authors would also discuss how multiple lines of systemic therapy delimits the efficacy of combination treatments.  

The reviewer suggests the authors to edit the tables to follow a single style. For example, in table 1, primary tumor site, the authors have provided % within brackets but in rest of the place no % within brackets. These minor mistakes have to be taken care.

Author Response

Reviewer 1

  • In a previous study presented in the 2019 ASCO meeting, the authors have studied the efficacy of the same combination in advance HNSCC in a phase II trial(https://ascopubs.org/doi/abs/10.1200/JCO.2019.37.15_suppl.6025). The authors in that presentation concluded that CDX-3379 in combination with cetuximab is well tolerated with the primary toxicity of diarrhea. Also, it was claimed that antitumor activity was observed in cetuximab-resistant HNSCC patients, including an ongoing, durable complete response. The authors need to elaborate the present study with the findings of 2019 in the discussion section as both the study has the same clinical trial initiative NCT03254927.

Response: Thank you for the opportunity to clarify. The 2019 ASCO poster, referenced in our manuscript (and in which numerous authors overlap), presented genomic results from 3 trials across the CDX-3379 development program where patients with HNSCC had been treated; this included 14 of 15 patients in first stage of the present phase II study. We have now clarified this within both the methods and discussion sections of the manuscript.

  • The authors are advised to include a paragraph in the discussion why studies are to be performed and to consider development of ErbB3 and EGFR inhibition in an earlier line of palliative therapy. The authors would also discuss how multiple lines of systemic therapy delimits the efficacy of combination treatments.  

Response: Thank you for this suggestion to strengthen the discussion. We have added two references describing the molecular mechanisms of resistance in HNSCC and linked this to our suggestion that future studies should consider an earlier line of therapy.

  • The reviewer suggests the authors to edit the tables to follow a single style. For example, in table 1, primary tumor site, the authors have provided % within brackets but in rest of the place no % within brackets. These minor mistakes have to be taken care.

Response: Thank you for noting these inconsistencies, which have been corrected in this revision.

Reviewer 2 Report

  • line, The optimal patient selection and de-escalation strategy is critically important in the evolving treatment of locoregional head and neck cancer. Several potential biomarkers to select patients for treatment de-escalation including clinical risk stratification have been assessed. Moreover, adjuvant de-escalation based on pathologic features, response to induction therapy, and molecular markers are all linked, permitting to optimize the management of the disease. doi:10.1634/theoncologist.2020-0303
  • line 66, Trans Oral Robotic Surgery (TORS) is a modality in the management of oropharyngeal squamous cell carcinoma (OPSCC), even in advanced stages such as T3, allowing tumor debulking and deintensification of postoperative treatment. Although the indication should be accurately discussed by the multidisciplinary tumor board, the SND seems to be effective in a TORS framework. please cite doi:10.1016/j.anl.2021.05.007
  • line 116, EGFR signaling confers resistance to radiotherapy and is a validated target in head and neck squamous cell carcinoma (HNSCC). The inhibition of EGFR in combination with radiotherapy improves local control and overall survival in these patients; however, therapeutic resistance limits the efficacy of this approach. Though clonal isolation of carcinoma cells exposed to increasing concentrations of cetuximab, an interesting paper found how resistant cells upregulate prosurvival ErbB3 and AKT signaling. Using EFM-19 cells and confirmatory analysis of protein levels, the authors demonstrate that cetuximab resistance is characterized by enhanced neuregulin expression identifying a novel adaptive mechanism of therapeutic resistance. Inhibition of this autocrine loop with CDX-3379 (an ErbB3 specific antibody) was sufficient to block ErbB3/AKT signaling in cetuximab resistant cells. The combination of CDX-3379 and cetuximab reduced proliferation and survival after radiotherapy in several HNSCC cell lines. Autocrine NRG ligand secretion is a mechanism for therapeutic resistance to cetuximab and radiotherapy. This cross-resistance to both therapeutic modalities identifies NRG as an actionable therapeutic target for improving treatment regimens in HNSCC. please cite doi:10.1158/1078-0432.CCR-18-3453

Methods

  • Toxicity was assessed through a score or a questionnaire?
  • please specify also in this article the biomarkers analysis method

Results

modify the flow diagram in consort model

Discussion

- line 303, NRG1 rearrangements are oncogenic drivers that are enriched in invasive mucinous adenocarcinomas (IMA) of the lung. The oncoprotein binds ERBB3-ERBB2 heterodimers and activates downstream signaling, supporting a therapeutic paradigm of ERBB3/ERBB2 inhibition. As proof of concept, a durable response was achieved with anti-ERBB3 mAb therapy (GSK2849330) in an exceptional responder with an NRG1-rearranged IMA on a phase I trial (NCT01966445). In contrast, response was not achieved with anti-ERBB2 therapy (afatinib) in four patients with NRG1-rearranged IMA (including the index patient post-GSK2849330). Although in vitro data supported the use of either ERBB3 or ERBB2 inhibition, these clinical results were consistent with more profound antitumor activity and downstream signaling inhibition with anti-ERBB3 versus anti-ERBB2 therapy in an NRG1-rearranged patient-derived xenograft model. Analysis of 8,984 and 17,485 tumors in The Cancer Genome Atlas and MSK-IMPACT datasets, respectively, identified NRG1 rearrangements with novel fusion partners in multiple histologies, including breast, head and neck, renal, lung, ovarian, pancreatic, prostate, and uterine cancers. please cite doi:10.1158/2159-8290.CD-17-1004

Author Response

Reviewer 2

  • line, The optimal patient selection and de-escalation strategy is critically important in the evolving treatment of locoregional head and neck cancer. Several potential biomarkers to select patients for treatment de-escalation including clinical risk stratification have been assessed. Moreover, adjuvant de-escalation based on pathologic features, response to induction therapy, and molecular markers are all linked, permitting to optimize the management of the disease. doi:10.1634/theoncologist.2020-0303
  • line 66, Trans Oral Robotic Surgery (TORS) is a modality in the management of oropharyngeal squamous cell carcinoma (OPSCC), even in advanced stages such as T3, allowing tumor debulking and deintensification of postoperative treatment. Although the indication should be accurately discussed by the multidisciplinary tumor board, the SND seems to be effective in a TORS framework. please cite doi:10.1016/j.anl.2021.05.007

Response: Thank you for your expert comments. The authors agree that both of these statements are true regarding the evolving treatment paradigms for locoregional HNSCC, including the use of biomarkers and risk classification systems for treatment intensification/de-intensification as well as the application of TORS and selective neck dissection in appropriately selected candidates. The presented manuscript reports on a phase II clinical trial in patients with recurrent/metastatic HNSCC. The purpose of the introduction is to frame the rates of failure following locoregional treatment, then hone in on the poor prognosis and lack of therapeutic options for those with refractory, HPV-negative, recurrent/metastatic disease. Adding additional introductory information on optimal management of locoregional disease would be out of place here and the authors respectfully disagree with adding these two references.

  • line 116, EGFR signaling confers resistance to radiotherapy and is a validated target in head and neck squamous cell carcinoma (HNSCC). The inhibition of EGFR in combination with radiotherapy improves local control and overall survival in these patients; however, therapeutic resistance limits the efficacy of this approach. Though clonal isolation of carcinoma cells exposed to increasing concentrations of cetuximab, an interesting paper found how resistant cells upregulate prosurvival ErbB3 and AKT signaling. Using EFM-19 cells and confirmatory analysis of protein levels, the authors demonstrate that cetuximab resistance is characterized by enhanced neuregulin expression identifying a novel adaptive mechanism of therapeutic resistance. Inhibition of this autocrine loop with CDX-3379 (an ErbB3 specific antibody) was sufficient to block ErbB3/AKT signaling in cetuximab resistant cells. The combination of CDX-3379 and cetuximab reduced proliferation and survival after radiotherapy in several HNSCC cell lines. Autocrine NRG ligand secretion is a mechanism for therapeutic resistance to cetuximab and radiotherapy. This cross-resistance to both therapeutic modalities identifies NRG as an actionable therapeutic target for improving treatment regimens in HNSCC. please cite doi:10.1158/1078-0432.CCR-18-3453

Response: The authors agree with the importance of neuregulin expression as an adaptive mechanism of therapeutic resistance to cetuximab. We have enhanced the text in the introduction section associated with references 11-14 (the preclinical papers, including the one described above which is cited as #12). Please also note that we already have included the additional suggested reference, doi:10.1158/1078-0432.CCR-18-3453, which is our prior window of opportunity study of CDX-3379 in operable HNSCC (Duvvuri U, … Bauman JE, Clin Cancer Res 2019).

Methods

  • Toxicity was assessed through a score or a questionnaire?

Response: Toxicity was assessed by the local treating investigator clinically and scored using the U.S. National Cancer Institute’s Common Toxicity Criteria for Adverse Events (CTCAE v.5). We have clarified this in the methods section.

  • please specify also in this article the biomarkers analysis method

Response: We have updated the Biomarker Analysis section and removed the “as previously described” statement.

Results

  • modify the flow diagram in consort model

Response: The authors are unclear on how the reviewer recommends modifying the flow diagram in the CONSORT model. In this revision, we have reviewed and updated it according to current CONSORT recommendations, making sure that formatting was consistent.

Discussion

  • line 303, NRG1 rearrangements are oncogenic drivers that are enriched in invasive mucinous adenocarcinomas (IMA) of the lung. The oncoprotein binds ERBB3-ERBB2 heterodimers and activates downstream signaling, supporting a therapeutic paradigm of ERBB3/ERBB2 inhibition. As proof of concept, a durable response was achieved with anti-ERBB3 mAb therapy (GSK2849330) in an exceptional responder with an NRG1-rearranged IMA on a phase I trial (NCT01966445). In contrast, response was not achieved with anti-ERBB2 therapy (afatinib) in four patients with NRG1-rearranged IMA (including the index patient post-GSK2849330). Although in vitro data supported the use of either ERBB3 or ERBB2 inhibition, these clinical results were consistent with more profound antitumor activity and downstream signaling inhibition with anti-ERBB3 versus anti-ERBB2 therapy in an NRG1-rearranged patient-derived xenograft model. Analysis of 8,984 and 17,485 tumors in The Cancer Genome Atlas and MSK-IMPACT datasets, respectively, identified NRG1 rearrangements with novel fusion partners in multiple histologies, including breast, head and neck, renal, lung, ovarian, pancreatic, prostate, and uterine cancers. please cite doi:10.1158/2159-8290.CD-17-1004

Response: Thank you for suggesting the inclusion of this important reference, which does enhance the discussion. Although only present in 0.5% of HNSCC in TCGA, activating NRG1 fusions may be a predictive biomarker for anti-ErbB3 drugs. We have added the reference and additional text accordingly.